# Accelerometry-Workload Indices Concerning Different Levels of Participation during Congested Fixture Periods in Professional Soccer: A Pilot Study Conducted over a Full Season

**DOI:** 10.3390/ijerph18031137

**Published:** 2021-01-28

**Authors:** Filipe Manuel Clemente, Rui Silva, Yung-Sheng Chen, Rodrigo Aquino, Gibson Moreira Praça, Julen Castellano, Hadi Nobari, Bruno Mendes, Thomas Rosemann, Beat Knechtle

**Affiliations:** 1Escola Superior Desporto e Lazer, Instituto Politécnico de Viana do Castelo, Rua Escola Industrial e Comercial de Nun’Álvares, 4900-347 Viana do Castelo, Portugal; filipe.clemente5@gmail.com (F.M.C.); rui.s@ipvc.pt (R.S.); 2Instituto de Telecomunicações, Delegação da Covilhã, 1049-001 Lisboa, Portugal; 3Department of Exercise and Health Sciences, University of Taipei, Taipei 11153, Taiwan; yschen@utaipei.edu.tw; 4Center of Physical Education and Sports, Department of Sports, Federal University of Espírito Santo, Vitoria 29075-910, Brazil; aquino.rlq@gmail.com; 5Sports Department, Universidade Federal de Minas Gerais, Belo Horizonte 31270-901, Brazil; gibson_moreira@yahoo.com.br; 6Department of Physical Education and Sport, University of Basque Country UPV EHU, 48940 Vitoria, Spain; julen.castellano@ehu.eus; 7Department of Exercise Physiology, Faculty of Sport Sciences, University of Isfahan, Isfahan 81746-73441, Iran; hadi.nobari1@gmail.com; 8Sports Scientist, Sepahan Football Club, Isfahan 81346-13119, Iran; 9Faculty of Human Kinetics, University of Lisboa, 1649-004 Lisboa, Portugal; brunomendes941982@gmail.com; 10Institute of Primary Care, University of Zurich, 8091 Zurich, Switzerland; thomas.rosemann@usz.ch; 11Medbase St. Gallen Am Vadianplatz, Vadianstrasse 26, 9001 St. Gallen, Switzerland

**Keywords:** association football, performance, GPS, external load, load monitoring, sports science

## Abstract

The aim of this study was to analyze the variations of acute load (AL), acute: chronic workload ratio (ACWR), training monotony (TM), and training strain (TS) of accelerometry-based GPS measures in players who started in three matches (S3M), two matches (S2M), and one match (S1M) during congested weeks. Nineteen elite professional male players from a Portuguese team (age: 26.5 ± 4.3 years) were monitored daily using global positioning systems (GPSs) over a full season (45 weeks). Accelerometry-derived measures of high metabolic load distance (HMLD), high accelerations (HA), and high decelerations (HD) were collected during each training session and match. Seven congested weeks were classified throughout the season, and the participation of each player in matches played during these weeks was codified. The workload indices of AL (classified as ACWR, TM, and TS) were calculated weekly for each player. The AL of HMLD was significantly greater for S2M than S1M (difference = 42%; *p* = 0.002; d = 0.977) and for S3M than S1M (difference = 44%; *p* = 0.001; d = 1.231). Similarly, the AL of HA was significantly greater for S2M than S1M (difference = 25%; *p* = 0.023; d = 0.735). The TM of HD was significantly greater for S2M than S3M (difference = 25%; *p* = 0.002; d = 0.774). Accelerometry-based measures were dependent on congested fixtures. S2M had the greatest TS values, while S3M had the greatest TM.

## 1. Introduction

The individualization of the training process requires, among other things, systematic monitoring of the load that occurs during sessions and competitions [1]. This kind of monitoring can be of paramount importance, especially considering that in team sports, the heterogeneity of the impact of exercises in players can be high [2], and there can be a discrepancy between coaches’ perceptions about the load imposed and the real impact imposed on players [3]. Thus, proper training load monitoring can help coaches to better adjust their training plans to the players and speed up the process of regulating stimuli to improve the recovery mechanisms of players [4].

Aspects of training load monitoring are commonly organized into two dimensions [5]: (i) external load, which is associated with the physical demands imposed on players and the mechanical work performed by players during the exercise; and (ii) internal load, which is related to the psychobiological effects of external load on the players. These two types of load are different in terms of the information available to sports scientists, even though they interact with each other and are related [6]. Internal load is commonly measured using heart rate monitors or rate of perceived exertion scales, while external load is quantified using devices such as global positioning systems (GPSs), accelerometers, and inertial measurement units [7]. 

Specific indicators of external load that are commonly measured using GPSs include: (i) distances covered at different speed thresholds; (ii) events associated with changes in speed, namely, accelerations/decelerations or changes in direction; and (iii) events related to the use of accelerometers or inertial measurement units (e.g., player load, impacts, or stride variables) [8]. The first two types of indicators (distances and changes in speed) can be highly variable in terms of tactical issues, while indicators of the third type tend to depend on the dynamics of the game [8]. Additionally, accelerometry-based measures can provide a great level of sensitivity and accuracy, considering the capacity of these sensors to collect data at a higher acquisition frequency than GPSs [7].

The quantification of acute load during training sessions and matches is important. However, a proper understanding of accumulated load can be crucial to identifying patterns in the training process and guaranteeing the correct progression and management of the load imposed across weeks [9]. In particular, the relationship between weekly load and chronic load (referred to as “acute: chronic workload ratio”, or ACWR) has been used to determine the progression and variation of load across weeks and to identify possible exposures to spikes in load [10]. Fundamentally, ACWR is a measure that can control the progression of load and quickly determine possible drastic and unplanned decreases and increases that may interfere with recovery/readiness and performance or affect injury risk [11,12]. Other indices such as training monotony and training strain can also be useful for monitoring load variations within a week and exposure to consistent high-doses, for example [13]. In particular, training monotony can provide information about the within-week variability of the load, while training strain indicates the overall impact of training on players [13]. These indices have also been used to determine possible relationships between bad overreaching, overtraining, and injury risk [14].

The variations of load can be planned by the coach or influenced by the competitive calendar. In fact, in team sports like soccer, seasons have become more congested, involving more periods of matches with few days of recovery in between [15]. Thus, scientific interest in congested fixture periods has increased in recent years, mainly considering the impact of congested periods on players’ performance [16], recovery processes [17], and injury risk [18]. Some of the possible risks of exposing players to congested fixture periods are reduced muscle stiffness [19], increased physiological stress and muscle damage [17], and greater strength deficits [20]. 

Extensive information regarding the acute impact of congested fixture periods has been provided in the last decade. Despite this, there is a lack of evidence about the influence of such periods on ACWR, training monotony, and strain. It is expected that these indices vary significantly in congested periods. Furthermore, it is possible that the heterogeneity of the indices between players increases. Differences in terms of workload indices between different levels of participation in matches could also be present—specifically between players who are starters (who begin the game) in three matches, two matches, or just one match in the same week. These possibilities must be described to improve our understanding of the impact of congested fixture periods on workload indices variations between players.

On the basis of the reasons stated above, and in an attempt to better characterize the impact of congested fixture periods on accelerometry-based indices regarding different levels of participation in matches, this study aimed to analyze variations of acute load, ACWR, training monotony, and the training strain of accelerometry-based GPS measures in starters of three matches (S3M), two matches (S2M), and one match (S1M) in professional soccer during congested weeks.

## 2. Materials and Methods

### 2.1. Experimental Approach to the Problem

This study used a cohort design. Throughout an entire 45-week season (3 July 2018 to 9 May 2019), the external loads of 19 professional soccer players were monitored daily, during both training sessions and matches. Weeks were classified as regular (one match per week) or congested (two matches or more within seven days). Considering the main purpose of this study (to compare workload indices between starters and non-starters in congested weeks), Table 1 presents the characteristics of the congested weeks included in the analysis. Considering the influence of matches on workload indices, the classification of players during congested weeks met the following criteria: (i) starters in three matches (S3M) who participated in three matches in the same week (for at least 45 min in each match); (ii) starters in two matches (S2M) who participated in two matches in the same week (playing for at least 45 min in each match); and (iii) starters in one match (S1M) who participated in a single match in a week (for at least 45 min of the game). 

The external loads of players were monitored daily using an 18-Hz GPS. The following accelerometry-derived measures were collected: (i) high metabolic power distance, and (ii) high accelerations and decelerations. Using the GPS measures, the weekly acute load, chronic load, acute: chronic workload ratio, training monotony, and training strain were calculated weekly. Information about the calculus of these outcomes can be found in the external load quantification section.

### 2.2. Participants

This study analyzed 19 professional men players (26.5 ± 4.3 years old; 75.6 ± 9.6 kg; 180.2 ± 7.3 cm; 7.5 ± 4.3 years of experience) belonging to a Portuguese European First League team. Among the participants, three were external defenders, four were central defenders, six were midfielders, four were wingers, and two were strikers. The inclusion criteria consisted of the following: (i) classified starters participated in at least 50% of the matches and 90% of training sessions in the three weeks before each analyzed congested week; (ii) none of the players were injured or ill in the congested weeks and in the three weeks preceding them; and (iii) none of the players were injured for more than four consecutive weeks during the entire season. A preliminary introduction to the study design and experimental approach was presented to the players. After their agreement, they signed a free written consent. The ethical standards of the Declaration of Helsinki for the study in humans were followed. 

### 2.3. External Load Quantification

Players were daily monitored with an 18-Hz GPS unit integrating a 100-Hz gyroscope, 100-Hz tri-axial accelerometer, and 10-Hz magnetometer (STATSports, Apex, Newry, Northern Ireland). The GPS revealed good validity and reliability levels [21,22]. An exclusive GPS unit was attributed to each player during the season, aiming to reduce the interunit variability. Players wore a specific vest with a bag placed on the upper back (interscapular line, T2–T4 vertebrae) with the GPS unit positioned inside. During data collection, the number of satellites varied between 18 and 21. The data collected in training sessions and matches were uploaded and treated in the STATSport Apex software (version 5.0).

The accelerometry-based measures collected daily were: (i) high metabolic load distance (high metabolic load distance (HMLD): corresponding to the distance covered at a speed greater than 5.5 m/s and while accelerating/decelerating at a magnitude of 2 m/s^2^ or above); (ii) high accelerations and decelerations (high accelerations (HA) and high decelerations (HD): the number of accelerations and decelerations with a magnitude of 3 m/s^2^ or above maintained for at least half of a second). The volume (total meters or number in each session) of each external load measure (during the session or match) was collected first for each player. After that, and for each subsequent week, the following indices were calculated: (i) acute load (wAL: corresponding to the sum of the load during a week); (ii) acute: chronic workload ratio (ACWL: representing the division of the wAL by the rolling average of accumulated training load in the previous four weeks—coupled version) [23]; (iii) training monotony (TM: corresponding to the mean of training load during the seven days of the week divided by the standard deviation of the days); (iv) training strain (TS: the multiplication of wAL by the TM) (Foster et al., 2001). These indices were calculated for each accelerometry-derived measure, resulting in the following variables: (i) wHMLD (weekly HMLD); (ii) acwrHMLD (ACWR of HMLD); (iii) mHMLD (monotony HMLD); (iv) sHMLD (strain HMLD); (v) wHA (weekly HA); (vi) acwrHA (ACWR of HA); (vii) mHA (monotony HA); (viii) sHA (strain HA); (ix) wHD (weekly HD); (x) acwrHD (ACWR of HD); (xi) mHD (monotony HD); and (xii) sHD (strain HD).

### 2.4. Statistical Procedures

The normality of the sample was assumed based on the central limit theorem, after being tested with the Kolmogorov–Smirnov test. Data were tested for his homogeneity using the Levene (*p* > 0.05). Descriptive statistics were presented in the form of tables and figures reporting the mean and standard deviation. The analysis of variation of the workload measures between types of participation was executed using mixed ANOVA. Tukey was used for pairwise comparisons since the sample was greater than 30. Statistical analysis was executed in the SPSS software (version 25.0, IBM, Chicago, IL, USA) for a *p* < 0.05. Effect size (ES) calculation was made following the Cohen’s approach (d) for a 95% confidence interval (95% CI). The magnitude of changes was interpreted based on the following thresholds [24]: 0.00 to 0.19, trivial; 0.20 to 0.59, small; 0.60 to 1.19, moderate; 1.20 to 1.99, large; >2.00, very large.

## 3. Results

Meaningful differences were found between type of participation in the congested weeks for the accelerometry-based workload measures (Table 2). The aHMLD was meaningfully greater for S2M than S1M (42%) and was greater for S3M than S1M (44%). Additionally, the mHLMD was meaningfully greater for S3M than S2M (14%). Finally, the sHMLD was meaningfully greater for S2M than S1M (41%).

Table 3 presents the differences between S1M, S2M, and S3M for aHA, acwrHA, mHA, and sHA. The aHA (25%), mHA (33%), and sHA (44%) were meaningfully greater for S2M than S1M.

The analysis of variation for aHD, acwrHD, mHD, and sHD can be observed in Table 4. The mHD was meaningfully greater for S2M than S3M (25%).

## 4. Discussion

This study aimed to analyze the variations of AL, ACWR, TM, and TS for accelerometry-based GPS measures at different levels of match participation among professional soccer players. The main evidence indicates that there are no significant differences between S1M, S2M, and S3M for ACWR for all measures. Meanwhile, S2M and S3M had greater ALs than S1M for all accelerometry-based measures.

Considering HMLD, it was found that S3M presented the greatest ALs. Additionally, S2M had the greatest TM and TS, while no significant differences were found for ACWR. HMLD is measured by the amount of high-speed running performed, combined with acceleration and deceleration distances [25]. This variable seems to be position-dependent [26,27]. Although there is a lack of evidence on the effects of accelerometry measures in congested periods [28], it has been demonstrated that distance-based measures are not dependent on congested periods [29,30]. This contrasts with our findings, in which weekly accumulated HMLD ALs seemed to be affected by congestion fixtures.

In a study conducted on 28 elite soccer players, it was found that in a regular week, players reached ~6000 m of HMLD [31], which is in line with the S1M group of our study. However, for players who started in two or three matches, HMLD reached 9809 m. As training sessions may be reduced (2–3 sessions) and are mainly related to recovery training sessions between matches in congested weeks [15], load variability was expected, which was reflected in the lower TM found in this study. Despite the lower TM and balanced ACWR found throughout the congested period, coaches should be aware of high strain values, such as those found in the S2M group (8328 A.U.), as it has been reported that the lower threshold that favors illness is ~6000 A.U. [32].

For HA comparisons, our results showed that S2M had greater AL, TM, and TS values than S1M and S3M, while no significant differences were found for ACWR. Arruda et al. [28] found that the number of accelerations decreased after a congested fixture. However, the authors did not consider HA separately, which might have skewed the results. In contrast, our study showed that S2M and S3M covered greater HA distances than S1M. These comparisons must be analyzed with caution since the authors of the aforementioned study did not distinguish starters from non-starters, despite using players with an expert level. In fact, in shorter periods (i.e., during a match), it was reported that HA decreased due to fatigue during the final minutes of the match [33,34]. However, the same trend was not observed in longer periods (i.e., accumulated matches), in which the ALs of HA increased significantly, mainly in S2M. Although fatigue does not seem to negatively affect HA performance in congested weeks, attention should be given to augmented TS values. Increased metabolic loads (due to the high metabolic demands of HA) and accumulated loads may increase the risk of injury [35,36,37]. 

Regarding the HD measure, it was found that S2M presented significantly greater TS values than S3M, while no significant differences were found for AL, ACWR, or TS between S1M, S2M, and S3M. Previous research showed that soccer teams complete greater HD than HA during matches than during training sessions [38]. However, there is a lack of research regarding weekly variations in congested fixtures, namely, considering important information such as the influence of playing position. Interestingly, in the present study, HD was lower than HA, both in regular and congested weeks. HD is closely related to mechanical work [39] and loading cycles are related to mechanical fatigue due to accumulated workloads [40]. However, recent research has warned that contrary to the damage caused by successive loads of mechanical work (such as HD), it is expected that human tissue is likely to cope with mechanical loads and augmented TS when applied for short periods [41]. Indeed, in the present study, it seemed that S2M and S3M were not affected by mechanical fatigue as they were able to withstand greater ALs and TS of HD; however, further research is needed to investigate the likelihood of an injury occurring in these cases. This can give coaches new insights about augmented HD TLs as a protective factor against injuries.

Our study had some limitations. The most evident was related to the sample size, as only one team was analyzed. Another limitation was that we did not consider other accelerometry measures, such as impact and fatigue indexes, which could give more detailed information about mechanical effects. Finally, playing positions were not considered due to the small sample. Future studies may provide more detailed information about positional dependencies, as well as the likelihood of injury occurring with increased loading cycles of HD distances in congested fixtures. Providing information about an entire session is certainly one of the strengths of this study. This issue represents a huge effort in terms of assuring external validity of the data under the real demands of a competitive soccer schedule, since it seems impossible to replicate in controlled environments (for example, in the laboratory).

Considering that ALs and TS for the overall accelerometry-based measures are expected to increase in congested fixtures, it is important to consider the proper management and preparation for this event. This may include a progressive overload in previous weeks to achieve a minimal spike in the week in which such a load will occur (congested weeks), as well as to identify exercises that could mitigate exposure to such frequencies during training sessions between matches during the congested period.

## 5. Conclusions

In this study, accelerometry-based measures were dependent on congested fixtures. The S2M group had the greatest TS values, while S3M had the greatest TM. Interestingly, S1M, S2M, and S3M had greater HA than HD. No significant ACWR changes were found. For these reasons, coaches should consider the effects of mechanical work in starters and non-starters to ensure that athletes can withstand high-intensity biomechanical loading cycle demands.

## Figures and Tables

**Table 1 ijerph-18-01137-t001:** Characterization of congested weeks included in this study.

Variable	CW1	CW2	CW3	CW4	CW5	CW6	CW7
Month	August	September	December		January		February
Week of the season (*n*)	9	14	23	26	27	31	32
Regular weeks before (*n*)	2	2	5	2	0	2	0
Training sessions between matches (*n*)	2	2	2	2	3	3	2
S3M (*n*)	3	4	6	6	2	4	4
S2M (*n*)	6	4	2	3	8	4	6
S1M (*n*)	8	6	2	2	4	6	3

CW: congested week; S3M: starter in three matches; S2M: starter in both matches; S1M: starter in one match.

**Table 2 ijerph-18-01137-t002:** Descriptive and inferential statistics (mean ± SD) of high metabolic load distances workload indices in different levels of participation in matches.

Outcome	S1M (Mean ± SD)	S2M (Mean ± SD)	S3M (Mean ± SD)	*p*	ES
aHMLD (m)	6817 ± 2677	9694 ± 3080	9809 ± 2261	S1M vs. S2M: 0.002 * S1M vs. S3M: 0.001 * S2M vs. S3M: >0.999	S1M vs. S2M: −0.977 moderate ^¶^ S1M vs. S3M: −1.231 large ^#^ S2M vs. S3M: −0.042 trivial
acwrHMLD (A.U.)	1.0 ± 0.5	1.0 ± 0.3	1.0 ± 0.3	S1M vs. S2M: >0.999 S1M vs. S3M: >0.999 S2M vs. S3M: >0.999	S1M vs. S2M: 0.000 trivial S1M vs. S3M: 0.000 trivial S2M vs. S3M: 0.000 trivial
mHMLD (A.U.)	0.8 ± 0.2	0.8 ± 0.3	0.7 ± 0.1	S1M vs. S2M: >0.999 S1M vs. S3M: 0.128 S2M vs. S3M: 0.010 *	S1M vs. S2M: 0.000 trivial S1M vs. S3M: 0.687 moderate ^¶^ S2M vs. S3M: 0.438 small ^&^
sHMLD (A.U.)	5922 ± 3200	8328 ± 3680	6666 ± 2269	S1M vs. S2M: 0.033 * S1M vs. S3M: >0.999 S2M vs. S3M: 0.130	S1M vs. S2M: −0.684 moderate ^¶^ S1M vs. S3M: −0.2789 small ^&^ S2M vs. S3M: 0.535 small ^&^

aHMLD: weekly acute load of high metabolic load distance; acwrHMLD: acute: chronic workload ratio of total distance; mHMLD: training monotony of total distance; sHMLD: training strain of total distance; S1M: starter in one match; S2M: starter in two matches; S3M: starter in three matches; *: *p*-value < 0.05; ^&^: small ES; ^¶^: moderate ES; ^#^: large ES; ES: effect size (standardized effect size of Cohen: d).

**Table 3 ijerph-18-01137-t003:** Descriptive and inferential statistics (mean ± SD) of high accelerations workload indices in different levels of participation in matches.

Outcome	S1M (Mean ± SD)	S2M (Mean ± SD)	S3M (Mean ± SD)	*p*	ES
aHA (m)	1134 ± 374	1423 ± 403	1348 ± 282	S1M vs. S2M: 0.023 * S1M vs. S3M: 0.155 S2M vs. S3M: >0.999	S1M vs. S2M: −0.735 moderate ^¶^ S1M vs. S3M: −0.667 moderate ^¶^ S2M vs. S3M: 0.213 small ^&^
acwrHA (A.U.)	1.1 ± 0.5	1.0 ± 0.2	1.0 ± 0.3	S1M vs. S2M: >0.999 S1M vs. S3M: >0.999 S2M vs. S3M: >0.999	S1M vs. S2M: 0.300 small ^&^ S1M vs. S3M: 0.255 small ^&^ S2M vs. S3M: 0.000 trivial
mHA (A.U.)	1.1 ± 0.4	1.2 ± 0.4	0.9 ± 0.2	S1M vs. S2M: 0.323 S1M vs. S3M: 0.187 S2M vs. S3M: <0.001 *	S1M vs. S2M: −0.250 small ^&^ S1M vs. S3M: 0.681 moderate ^¶^ S2M vs. S3M: 0.930 moderate ^¶^
sHA (A.U.)	1274 ± 734	1752 ± 809	1213 ± 448	S1M vs. S2M: 0.060 S1M vs. S3M: >0.999 S2M vs. S3M: 0.009 *	S1M vs. S2M: −0.610 moderate ^¶^ S1M vs. S3M: 0.106 trivial S2M vs. S3M: 0.810 moderate ^¶^

aHA: the weekly acute load of high accelerations; acwrHA: acute: chronic workload ratio of high accelerations; mHA: training monotony of high accelerations; sHA: training strain of high accelerations; S1M: starter in one match; S2M: starter in two matches; S3M: starter in three matches; *: *p*-value < 0.05; ^&^: small ES; ^¶^: moderate ES; ES: effect size (standardized effect size of Cohen: d).

**Table 4 ijerph-18-01137-t004:** Descriptive and inferential statistics (mean ± SD) of high decelerations workload indices in different levels of participation in matches.

Outcome	S1M (Mean ± SD)	S2M (Mean ± SD)	S3M (Mean ± SD)	*p*	ES
aHD (m)	966 ± 343	1201 ± 370	1166 ± 277	S1M vs. S2M: 0.057 S1M vs. S3M: 0.151 S2M vs. S3M: >0.999	S1M vs. S2M: −0.652 moderate ^¶^ S1M vs. S3M: −0.658 moderate ^¶^ S2M vs. S3M: 0.106 trivial
acwrHD (A.U.)	1.0 ± 0.5	1.0 ± 0.2	1.0 ± 0.3	S1M vs. S2M: >0.999 S1M vs. S3M: >0.999 S2M vs. S3M: >0.999	S1M vs. S2M: 0.000 trivial S1M vs. S3M: 0.000 trivial S2M vs. S3M: 0.000 trivial
mHD (A.U.)	0.9 ± 0.3	1.0 ± 0.3	0.8 ± 0.2	S1M vs. S2M: 0.568 S1M vs. S3M: 0.268 S2M vs. S3M: 0.002 *	S1M vs. S2M: −0.333 small ^&^ S1M vs. S3M: 0.411 small ^&^ S2M vs. S3M: 0.774 moderate ^¶^
sHD (A.U.)	947 ± 546	1290 ± 670	956 ± 384	S1M vs. S2M: 0.116 S1M vs. S3M: >0.999 S2M vs. S3M: 0.067	S1M vs. S2M: −0.545 small ^&^ S1M vs. S3M: −0.020 trivial S2M vs. S3M: 0.601 moderate ^¶^

aHD: weekly acute load of high decelerations; acwrHD: acute: chronic workload ratio of high decelerations; mHD: training monotony of high decelerations; sHD: training strain of high decelerations; S1M: starter in one match; S2M: starter in two matches; S3M: starter in three matches; *: *p*-value < 0.05; ^&^: small ES; ^¶^: moderate ES; ES: effect size (standardized effect size of Cohen: d).

## Data Availability

Data available on request due to restrictions eg privacy or ethical.

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
