# Peer review of "Accelerometry-Workload Indices Concerning Different Levels of Participation during Congested Fixture Periods in Professional Soccer: A Pilot Study Conducted over a Full Season"

_ijerph, 2021, doi:10.3390/ijerph18031137_

Round 1

Reviewer 1 Report

Dear Authors, thank you for the paper. Please, find some comments and questions below.

I would like to add a pilot study on the topic of the paper due to the paper's limitations and the difficulty in generalizing the results.

Line 121

The external loads of players were monitored daily using an 18-Hz GPS. The following accelerometry-derived measures were collected: (i) high metabolic power distance and (ii) high accelerations and decelerations. Weekly acute load, chronic load, acute: chronic workload ratio, training monotony, and training strain were also calculated 15 weekly for each measure.

It seems for the readers that the GPS calculated the second part of the parameter!

Can you explain how did you calculate the highlighted part above? 

Line 138 all the players were not injured or ill during the periods of congested weeks and 138 the three weeks before them; (iii) all the players were not injured for more than four consecutive weeks during all season.

Line 86-90

Thus, scientific interest in congested fixture periods has increased in recent years, mainly considering the impact of congested periods on players’ performance [16], recovery processes [17], and injury risk [18]. Some of the possible risks of exposing players to congested fixture periods are reduced muscle stiffness [19], increased physiological stress and muscle damage [17], greater strength deficits [20].

Do you not think that there is a contradiction between this result and what was written in the introduction ''Line 86-90''        

Line 149

Players wore a specific vest with a bag placed between the scapula in which the GPS unit was positioned. During data collection, the number of satellites varied between....

Can you please provide a Figure explain this step? Do you think it affects the players' performance?

Line 151 the number of satellites varied between 150 18 to 21

What does the number express? Is it a distance error?

Line 244 These comparisons must be analyzed with caution since the authors of the aforementioned study did not distinguish starters from non-starters.

But in this study, I think we also cannot distinguish between beginners and non-beginners due to the age of the players: 26.5 ± 4.3 years old and they seem to have some experience of years.

Are you sure these players have not competed in competitions before?
What is the line between beginners and others?

Line 259

Yes, loading cycles are related to mechanical fatigue due to accumulated workloads but this is also related to the players' positions. As we cannot generalize the results without taking players' positions into account!
Did the authors notice differences between the players' positions as well?
It will be interesting to look at these results as well?

I think given the limitations of the paper and having adjusted it that might be fine for me.      

Author Response

REVIEWER 1

Dear Authors, thank you for the paper. Please, find some comments and questions below.

AUTHORS: DEAR REVIEWER, THANK YOU FOR ALL YOU SUGGESTIONS AND COMMENTS. WE HAVE FOLLOWED ALL OF THEM. CHANGES RELATED TO YOUR WORK WERE HIGHLIGHTED IN GREEN.

I would like to add a pilot study on the topic of the paper due to the paper's limitations and the difficulty in generalizing the results.

AUTHORS: DEAR REVIEWER THANK YOU. WE HAVE CHANGED THE SUB-TITLE TO “A pilot study conducted over a full-season”

Line 121

The external loads of players were monitored daily using an 18-Hz GPS. The following accelerometry-derived measures were collected: (i) high metabolic power distance and (ii) high accelerations and decelerations. Weekly acute load, chronic load, acute: chronic workload ratio, training monotony, and training strain were also calculated 15 weekly for each measure.

It seems for the readers that the GPS calculated the second part of the parameter!

Can you explain how did you calculate the highlighted part above? 

AUTHORS: DEAR REVIEWER THANK YOU. WE HAVE CHANGED TO “Using the GPS measures, the weekly acute load, chronic load, acute: chronic workload ratio, training monotony, and training strain were calculated weekly. Information about the calculus of these outcomes can be found in the external load quantification section.”

Line 138 all the players were not injured or ill during the periods of congested weeks and 138 the three weeks before them; (iii) all the players were not injured for more than four consecutive weeks during all season.

AUTHORS: DEAR REVIEWER, THANK YOU.

Line 86-90

Thus, scientific interest in congested fixture periods has increased in recent years, mainly considering the impact of congested periods on players’ performance [16], recovery processes [17], and injury risk [18]. Some of the possible risks of exposing players to congested fixture periods are reduced muscle stiffness [19], increased physiological stress and muscle damage [17], greater strength deficits [20].

Do you not think that there is a contradiction between this result and what was written in the introduction ''Line 86-90''        

AUTHORS: DEAR REVIEWER, THANK YOU. ACTUALLY, NOT. THE FACT IS THAT WE HAVE SHOWN THAT IN CONGESTED PERIODS, WORKLOAD WAS MEANINGFULLY INCREASED WHICH MAY CONDUCT TO A EXPOSURE TO INJURY RISK.

Line 149

Players wore a specific vest with a bag placed between the scapula in which the GPS unit was positioned. During data collection, the number of satellites varied between....

Can you please provide a Figure explain this step? Do you think it affects the players' performance?

AUTHORS: DEAR REVIEWER, THANK YOU. WE DO NOT HAVE A PHOTO. BUT WE HAVE CHANGED THE SPECIFIC PLACE: Players wore a specific vest with a bag placed on the upper back (interscapular line, T2-T4 vertebrae) in which the GPS unit was positioned.

Line 151 the number of satellites varied between 150 18 to 21

What does the number express? Is it a distance error?

AUTHORS: DEAR REVIEWER, THANK YOU. THIS IS JUST INFORMATION FOR REPRODUCIBILITY OF PROTOCOL.

Line 244 These comparisons must be analyzed with caution since the authors of the aforementioned study did not distinguish starters from non-starters.

But in this study, I think we also cannot distinguish between beginners and non-beginners due to the age of the players: 26.5 ± 4.3 years old and they seem to have some experience of years.

Are you sure these players have not competed in competitions before?
What is the line between beginners and others?

AUTHORS: DEAR REVIEWER, THANK YOU. WE HAVE CHANGED TO “These comparisons must be analyzed with caution since the authors of the aforementioned study did not distinguish starters from non-starters, despite using players with an expert level.

Line 259

Yes, loading cycles are related to mechanical fatigue due to accumulated workloads but this is also related to the players' positions. As we cannot generalize the results without taking players' positions into account!
Did the authors notice differences between the players' positions as well?
It will be interesting to look at these results as well?

I think given the limitations of the paper and having adjusted it that might be fine for me.    

 AUTHORS: DEAR REVIEWER, THANK YOU. WE HAVE CHANGED THE SENTENCE TO “However, there is a lack of research regarding weekly variations in congested fixtures, namely considering important information as the influence of playing position.” ADDITIONALLY, WE HAVE ADDED THIS AN OUR STUDY LIMITATION: “Finally, playing positions were not considered due to the small sample.”

Reviewer 2 Report

This article is well written and presents the results clearly and is easy to read for those who are familiarized with the terms. Although the article is good in its present form, there are some issues that author should address before final publication.

Line 28. Indicate the league where they play.

Line 132. Indicate the country.

Line 159. The following indices were calculated.

Line 173. What statistic was used to analyse the normality?

Line 177. Explain why you chose Tukey rather than other one.

Line 182. There are mistakes with the brackets. Please, revise

Author Response

REVIEWER 2

This article is well written and presents the results clearly and is easy to read for those who are familiarized with the terms. Although the article is good in its present form, there are some issues that author should address before final publication.

AUTHORS: DEAR REVIEWER, THANK YOU FOR ALL YOU SUGGESTIONS AND COMMENTS. WE HAVE FOLLOWED ALL OF THEM. CHANGES RELATED TO YOUR WORK WERE HIGHLIGHTED IN BLUE.

Line 28. Indicate the league where they play.

AUTHORS: DEAR REVIEWER, THANK YOU. WE HAVE ADDED THE COUNTRY (PORTUGAL)

Line 132. Indicate the country.

AUTHORS: DEAR REVIEWER, THANK YOU. WE HAVE ADDED THE COUNTRY (PORTUGAL)

Line 159. The following indices were calculated.

AUTHORS: DEAR REVIEWER, THANK YOU. WE HAVE CHANGED ACCORDINGLY.

Line 173. What statistic was used to analyse the normality?

AUTHORS: DEAR REVIEWER, THANK YOU. WE HAVE ADDED “after being tested with Kolmogorov-Smirnov test.”

Line 177. Explain why you chose Tukey rather than other one.

AUTHORS: DEAR REVIEWER, THANK YOU. WE HAVE ADDED “Tukey was used since the sample was greater than 30.”

Line 182. There are mistakes with the brackets. Please, revise

AUTHORS: DEAR REVIEWER, THANK YOU. WE HAVE CHANGED TO “0.00 to 0.19, trivial; 0.20 to 0.59, small; 0.60 to 1.19, moderate; 1.20 to 1.99, large; >2.00, very large.”

Round 2

Reviewer 1 Report

It seems clearly modified.